**Data Availability Statement:** All data and syntax files are available from the OSF database at the following URL: https://osf.io/n7z2v/.

**Funding:** The author(s) received no specific funding for this work.

# Quantifying the effect of Wakefield et al. (1998) on skepticism about MMR vaccine safety in the U.S.

**Matthew Motta**[1]*, **Dominik Stecula**[2]

**1** Department of Political Science, Oklahoma State University, Stillwater, OK, United States of America,
**2** Department of Political Science, Colorado State University, Fort Collins, CO, United States of America

* matthew.motta@okstate.edu

## Abstract

### Background

Efforts to trace the rise of childhood vaccine safety concerns in the US often suggest Andrew Wakefield and colleagues' retracted 1998 *Lancet* study (AW98)–which alleged that the MMR vaccine can cause children to develop autism–as a primary cause of US vaccine skepticism. However, a lack of public opinion data on MMR safety collected before/after AW98's publication obscures whether anecdotal accounts are indicative of a potentially-causal effect.

### Methods

We address this problem using a regression discontinuity framework to study change in monthly MMR injury claims (N = 74,850; from 1990–2019) from the *Vaccine Adverse Events Reporting System* (VAERS) to proxy concern about vaccine safety. Additionally, we suggest a potential mechanism for the effect of AW98 on vaccine skepticism, via automated sentiment analyses of MMR-related news stories (N = 674; from 1996–2000) in major television and newspaper outlets.

### Results

AW98 led to an immediate increase of about 70 MMR injury claims cases per month, averaging across six estimation strategies (meta-analytic effect = 70.44 [52.19, 88.75], p < 0.01). Preliminary evidence suggests that the volume of negative media attention to MMR increased in the weeks following AW98's publication, across four estimation strategies (meta-analytic effect = 9.59% [3.66, 15.51], p < 0.01).

### Conclusions

Vaccine skepticism increased following the publication of AW98, which was potentially made possible by increased negative media coverage of MMR.

**Competing interests:** The authors have declared that no competing interests exist.

## Significance

Childhood vaccine skepticism presents an important challenge to widespread vaccine uptake, and undermines support for pro-vaccine health policies. In addition to advancing our understanding of the previously-obscured origins of US vaccine skepticism, our work cautions that high-profile media attention to inaccurate scientific studies can undermine public confidence in vaccines. We conclude by offering several recommendations that researchers and health communicators might consider to detect and address future threats to vaccine confidence.

## Introduction

In late February 1998, a research team led by Andrew Wakefield published an article in *The Lancet* suggesting a link between the Measles, Mumps, and Rubella vaccine (MMR) and the development of autism in children. An investigation into the study (hereafter, AW98), conducted more than ten years after its original publication date, concluded that the article's data did not support this claim, and documented credible evidence of research malfeasance [1].

Although the piece was eventually retracted in Spring 2010 [1], both scholarly [2–4] and journalistic [5] efforts to identify the origins of contemporary vaccine skepticism in the U.S. suggest that the publication of AW98— and early media attention to it—was a pivotal moment (and perhaps *the* pivotal moment) in the mainstream acceptance of skepticism about MMR vaccine safety. Today, approximately one in three Americans believes that childhood vaccines can cause children to develop autism [6].

Studies attributing the rise in childhood vaccine skepticism to AW98, however, rely primarily on anecdotal evidence [2, 4–5]. This may be due in part to the scarcity of polling data about MMR safety in the immediate aftermath of AW98, and the virtual absence of such data prior to 1998 [7]. One way that scholars have circumvented this concern is by investigating the effect of AW98 on childhood vaccine skepticism is to assess whether or not MMR *refusals* (presumably resulting from safety concerns) increased in the aftermath [3]. Using this approach, AW98—and media coverage of it—appears to have had a short-lived and limited influence on vaccine refusal rates.

However, as outright MMR refusal is an extreme form of action—complicated by the fact that taking parents may be required to seek medical or philosophical exemptions to immunization statutes [8]—it may lack the sensitivity necessary to detect broader changes in public opinion toward MMR. In this study, make use of a comparatively more-sensitive indicator of MMR skepticism; parent reports of negative reactions to the MMR vaccine from the Vaccine Adverse Events Reporting System (VAERS; Department of Health & Human Services). These reports enable parents to express skepticism about MMR safety by reporting potential side effects their children may have experienced; meaning that parents need not take the more-extreme step of refusing vaccines outright to register concerns about its safety.

If VAERS reports are sufficiently-sensitive indicators of MMR skepticism, and if conventional wisdom about AW98 is correct, we should expect to see a sharp increase in reports following the publication of—and media attention to—the debunked study. While VAERS reports, unlike public opinion surveys, cannot tell us the "base rate" of MMR skeptical views in the mass public, they can at least offer insights into whether or not the levels of skepticism necessary to attribute MMR to various potential side effects following AW98.

Empirically demonstrating whether or not AW98 "moved the needle" on vaccine skepticism is an important empirical task, as it can help scholars better understand how communication surrounding debunked or misleading science might go on to shape behaviors relevant to public health. Additionally, given the link between MMR skepticism and opposition to policies that encourage vaccination [7, 9], this research can help us understand the potential long-run impacts of attention to fraudulent scientific claims on support for public health policies.

## Materials & methods

### Study design, population, and setting

Data from this study come from two sources. First, we test whether or not concern about MMR safety increased following the publication of AW98 using publicly available data of reports of adverse reactions to the MMR vaccine, via the Department of Health and Human Services (DHHS) Vaccine Adverse Event Reporting System (VAERS) [10]. Doctors and other vaccine providers report potential adverse events to VAERS in consultation with the parents of children administered the MMR vaccine.

Next, we provide supplemental tests of whether or not media coverage of AW98 presents a plausible mechanism for the effects documented in Fig 1. We obtained data from Lexis Nexis Academic by searching for news stories referencing "measles" or "MMR" in national television broadcasts (ABC, CBS, PBS, and NBC News) and high-circulation newspapers (USA Today, Washington Post, and the New York Times, Associated Press). We measured story sentiment using Linguistic Inquiry and Word Count (LIWC) software [11], which computes the ratio of

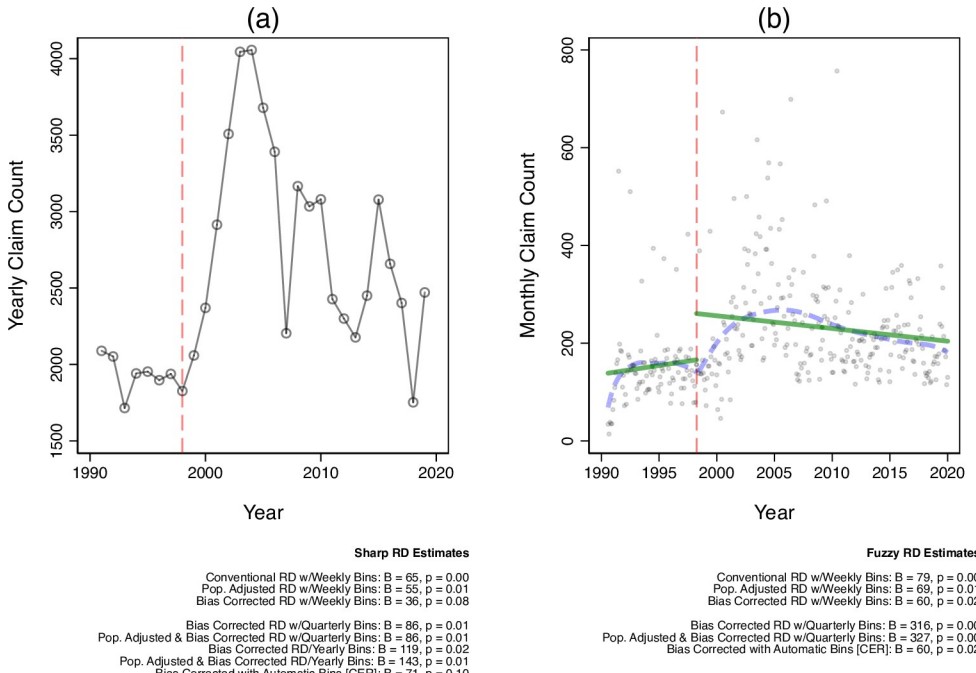

**Fig 1. MMR VAERS reports pre/post AW98.** Panel a presents yearly MMR VAERS reports before/after the publication of AW98 (dashed vertical line). Panel b presents RD estimates of the effect (B) and two-tailed significance (p) of AW98 on weekly report counts before/after AW98 (dashed vertical line), across several RD estimation strategies (see: Materials & Methods). All RD estimates are calculated as linear effects; which is appropriate given the high degree of consonance between the locally weighted polynomial trend line (dashed lines) and linear trend line (solid line) fit to the monthly data.

negatively to positively valenced words in each story, standardized on a scale ranging from 0 (most negative) to 100 (most positive).

## Outcome measures

Our measure of MMR safety concerns is a count of all monthly MMR event reports filed to VAERS from 1990 (when the program was created) to 2019. Note that while several varieties of MMR are in use today (e.g., MMRV, which includes varicella), we focus on *just* MMR; both because it was the subject of AW98, and because it is consistently available throughout the series.

Additionally, for our supplemental tests, negative news volume about the MMR vaccine is the average (mean) negativity score for all MMR-related stories produced each week from March 1996 (two years before AW98's publication) to March 2000 (two years post). We weighted weekly averages by the total number of stories featured in that week.

## Statistical analysis

To test whether or not AW98 had a causal effect on increased MMR safety concerns, we first compare the number of MMR reports filed to VAERS pre/post-AW98 using a regression discontinuity (RD) setup. For robustness, we present several versions of the RD results that vary (1) whether the RD is "sharp" (estimated before/after the date AW98 was published) or "fuzzy" (within a month of the paper's publication); (2) report aggregation level (monthly, quarterly, yearly, or using an automatic "bin" selection method via a coverage error rate [CER] optimal bandwidth estimator); (3) population adjusted vs. non-adjusted effect size estimators (to ensure that a potential increase in reports is not the result of population growth over time), and (4) the use of bias-correction standard error estimates. Results of all models are presented in Fig 1. All data and code necessary to replicate these analyses, as well as the placebo tests referenced above, can be found at: https://osf.io/n7z2v/.

To detect whether or not media coverage of AW98 might be responsible for increased MMR concern, we again use an RD setup; this time aggregating both the volume and tone of MMR stories from two years before and after the publication of AW98 (1996–2000). These tests, visualizations, and the time demarcations are constructed identically to those reported in Fig 1 —modeling change in sentiment over time—with the exception that: (1) we restrict data aggregation to be at the weekly level (as monthly aggregations would leave the model underpowered; daily aggregations would have 'sparse' days with no coverage); (2) all models include story counts as a covariate (as we are interested in account for the *volume* of negative coverage); and, (3) both the locally weighted and linear trend lines adjust for weekly story volume.

In both cases, we summarize the results of these estimation procedures with a formal meta-analysis, using Cohen's standardized mean difference procedure. Note that we report these quantities at only the monthly level for the adverse events data. This serves as a conservative estimate of the effect of AW98 on vaccine skepticism, as reporting the larger aggregation periods (included as robustness checks) would potentially inflate the average effect size.

## Results

Fig 1 presents an initial test of AW98's effect on MMR safety concerns. For reference, panel a presents the total yearly count of reported adverse experiences with MMR, from 1990 to 2019. Panel b visualizes the results of several regression discontinuity (RD) estimates of both the substantive effect of AW98 on monthly VAERS reports and statistical significance. If AW98 did indeed influence public perceptions of MMR vaccine safety, would we expect to see a

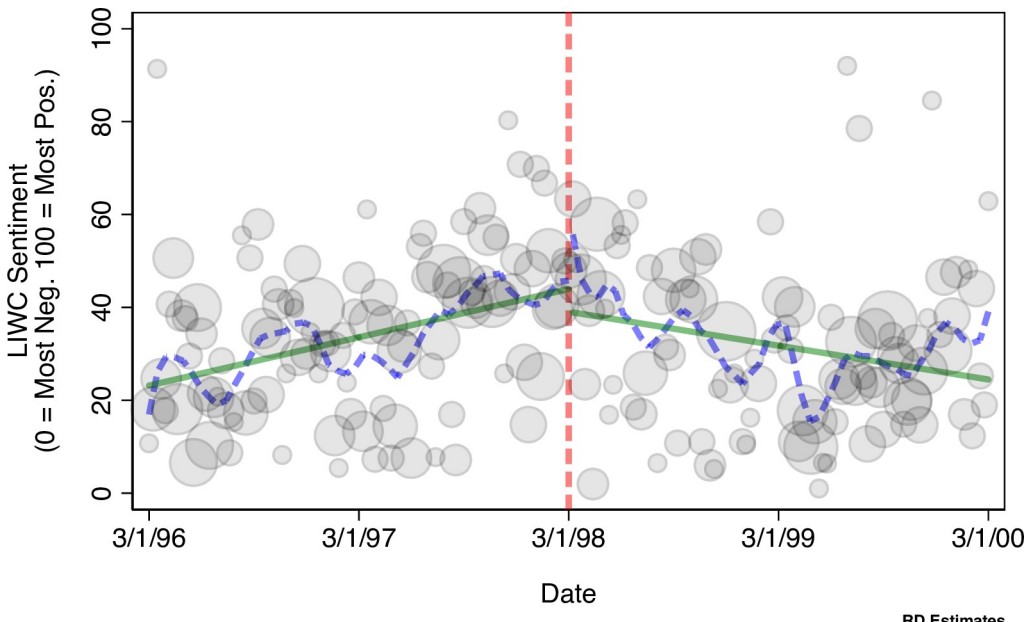

RD Estimates

Sharp Conventional RD w/Weekly Bins, Weighted by # of Stories: B = -9.07, p = 0.10
Fuzzy Conventional RD w/Weekly Bins, Weighted by # of Stories: B = -10.97, p = 0.10
Sharp Bias Corrected RD w/Weekly Bins, Weighted by # of Stories: B = -7.71, p = 0.16
Fuzzy Bias Corrected RD w/Weekly Bins, Weighted by # of Stories: B = -11.64, p = 0.08

**Fig 2. MMR news sentiment pre/post AW98 (weighted by volume).** Figure presents weekly sentiment ratings of vaccine-related stories from major news outlets (gray shaded circles), weighted by the total amount of stories published that week (larger when there is more coverage in a given week). The figure also reports the effect (B) and two-tailed significance (p) of AW98 on weekly report counts before/after AW98 (dashed vertical line), across several RD estimation strategies (see: Materials & Methods). All RD estimates are calculated as linear effects; which is appropriate given the high degree of consonance between the locally weighted polynomial trend line (dashed lines) and linear trend line (solid line) fit to the weekly data; both of which adjust for total volume. N = 674 stories.

substantively large and statistically significant increase in the period following the paper's publication, compared to the period before it.

The descriptive results presented in Fig 1A document that, prior to the publication of AW98, parents typically reported under 2,000 adverse MMR events per year. Following AW98, that quantity steeply grew to over 4,000 yearly reports by 2004. Formal RD tests presented in Fig 2B document a large and statistically significant increase in events immediately following the publication of AW98. Meta-analyzing the six monthly tests presented in the figure suggests an increase of 70.44 [52.19, 88.75] reports per week.

These results hold across tests that vary both the report aggregation period, standard error estimator, and whether AW98 is treated as a "sharp" or "fuzzy" discontinuity. The results also hold when accounting for population growth over time (which might result in more reports being filed).

Additionally, to ensure that growth MMR concern is not confounded by (1) changes in the procedures by which parents report adverse effects to VAERS over time (e.g., the ability to submit reports online as personal home computing and internet access grew over time), (2) changes in how health care providers diagnose autism, and/or (3) time itself–i.e., the possibility that attitudes toward all vaccines grew more negative over time for reasons unrelated to AW98 –we offer a placebo test using reports of adverse reactions to HIBV (Haemophilus Influenzae Type B Vaccine)–which has been routinely administered to children since the early 1990s, but was *not* identified as a health risk in AW98 –in the online materials.

If the effects of AW98 were to coincide with unobserved changes in reporting mechanisms and/or more general changes in public vaccine sentiment, we should observe comparatively more VAERS reports in the post (vs. pre) AW98 period for the HIBV placebo. This would be indicative of a spurious effect of AW98. However, the analyses presented in the online materials suggest that AW98 had no statistically or substantively discernable effect on HIBV adverse event reports (see: S1 Fig).

Fig 2 presents evidence of a potential mechanism for this effect. In the weeks following AW98's publication, the volume of negative news stories about the MMR vaccine from major news outlets increased substantially. Meta-analyzing the RD analyses presented in Fig 2 suggests that negativity increased by 9.59% [3.66, 15.51] following publication, approaching conventional two-tailed significance in three out of four estimation strategies. This provides preliminary evidence that AW98 influenced how the media talked about MMR, which in turn drew public attention to concerns about vaccine safety.

## Conclusions

This research demonstrates that AW98, and media attention to it, may have changed how some Americans viewed MMR safety. Whereas anecdotal accounts suggest that AW98 led to an increase in vaccine skepticism in the US, early studies of pre/post AW98 vaccine compliance rates cast doubt on the study's impact on public opinion. By constructing more sensitive indicators of vaccine skepticism, we detect a large, robust, and statistically significant uptick in public concern about MMR safety following AW98.

Without public opinion data, of course, we cannot determine precisely how many Americans came to hold negative views toward MMR post-AW98. However, our approach allows us to document *change* in vaccine sentiment over time attributable to AW98; enabling us to shed new light on a previously-muddled area of vaccine history.

## Discussion

Many Americans hold skeptical views about the safety of childhood vaccines [7, 9, 12–15]. Vaccine skepticism has important public health consequences, as people who believe that vaccines are unsafe tend to be less likely to intend to vaccinate themselves and their children against vaccine-preventable illnesses [14–16], and more likely to oppose pro-vaccine health policies [7, 17]. Understanding the origins of vaccine skepticism in the U.S. can help researchers better preempt how to mitigate the effect that fraudulent claims might have on public vaccine opinion in the future [18].

Consequently, this research has several important public health consequences. First, it suggests that attention to false or misleading vaccine research can impact public confidence in vaccines. Media sources should therefore work in consultation with researchers to stringently vet vaccine-related stories before sharing their results with the public. This will be particularly important in a post COVID-19 pandemic public health environment, where scholars have the opportunity to produce research on the (potential) long-term side effects of vaccines currently approved for public use.

Additionally, our work underscores the importance of conducting regular public opinion polling about vaccine safety; even *before* the possibility of controversy. While VAERS is a useful way to detect change in vaccine attitudes in the absence of public opinion data, surveys enable us to more-precisely document change in vaccine skepticism, over time. Constant surveillance of vaccine opinion can help researchers better understand how changes in the media environment might influence public vaccine confidence, and (potentially) thereby influence related vaccine policy attitudes and health behaviors.

Finally, our work suggests several opportunities for future research. For example, our study cannot disentangle the *precise* mechanism by which AW98 media coverage might influence public vaccine attitudes. Media attention could, for example, directly influence how Americans feel about vaccination by drawing parents' attentions to the possibility that their children experience adverse side effects from vaccination. Alternatively, the vaccine media environment could indirectly influence vaccine attitudes by changing health care professionals attentiveness to potential side effects (e.g., increased monitoring for autism symptoms [19]), and thereby engender concern among parents resulting in increased VAERS reports. Efforts to disentangle direct from more-complex causal accounts are a worthwhile endeavor in previous research.

Additionally, and perhaps most importantly, our research offers an opportunity to learn more about the effects of COVID-19 vaccine media coverage on attitudes toward vaccination. As more Americans became eligible to vaccinate against the virus, vaccine skeptics and prominent media figures who held skeptical views toward vaccination sometimes made use of VAERS reports to justify their (often, misinterpreted) concerns about COVID-19 vaccine safety [20–22]. Consequently, future work should make an effort to determine whether or not media coverage of these issues in turn influenced reports to VAERS. This information can help scholars to not only better understand the effects of media coverage on vaccine attitudes, in application to diseases not studied in our work, but how the quality of adverse effect data reported to VAERS might be influenced by exogenous media, political, and other social events.

Moreover, although public opinion data about COVID-19 vaccine attitudes and behaviors is comparatively more plentiful than MMR opinion data, few surveys ask uptake questions at the daily level. As we demonstrate in this research, VAERS data–which can be readily aggregated at the daily level –could function as a useful supplement for public opinion data to detect short-term effects of changes in the vaccine communication environment. For example, researchers could use VAERS data to document the effects of media attention the federal government's decision to temporarily pause administration of Johnson & Johnson's COVID-19 vaccine. Because this pause lasted just ten days [23], VAERS data may offer an opportunity to assess these effects with additional granularity.

## Supporting information

**S1 Fig. Placebo Test: HIBV adverse effect reports.** Placebo test replicates the analyses and tests presented in Fig 1 in the main text (although note that these analyses also account for exponential decline in VAERS reports following the publication of AW98 in the bias correction RD models by specifying a quadratic regression estimator). Please refer to the main text for additional methodological details. HIBV refers to the Haemophilus Influenzae Type B Vaccine, which–while typically administered in childhood since the early 1990s, like MMR—was not mentioned as a potential cause of autism in the AW98 paper. Consequently, if the effects documented in Fig 1 are merely the result of unobserved confounds, or time itself, we would expect to see a corresponding spike in adverse event reports following the publication of AW98. Consistent with the idea that the effects of AW98 are not confounded, the results document little effect of AW98 on VAERS reports for HIBV. Visually, although adverse event reports rise slightly following the publication of AW98, reports (1) were already increasing *prior to* its publication, and (2) had been in a period of exponential decline before 1998. Statistically, the results fail to document a significant increase in VAERS reports after correcting for the (pronounced) inverted parabolic nature of the pre-AW98 data, using quadratic regression. Note that because the analyses both substantively and statistically document no evidence of a discontinuity in adverse reports following AW98 in higher-powered (weekly) analyses, we do

not re-estimate at the yearly or quarterly level.
(TIF)

## Author Contributions

**Conceptualization:** Matthew Motta, Dominik Stecula.

**Data curation:** Matthew Motta, Dominik Stecula.

**Formal analysis:** Matthew Motta.

**Investigation:** Matthew Motta, Dominik Stecula.

**Methodology:** Matthew Motta.

**Project administration:** Matthew Motta.

**Resources:** Dominik Stecula.

**Validation:** Matthew Motta.

**Visualization:** Matthew Motta.

**Writing – original draft:** Matthew Motta, Dominik Stecula.

**Writing – review & editing:** Matthew Motta, Dominik Stecula.

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
