## [Decision Letter · Decision Letter 0]

21 Jul 2021

PONE-D-21-02863

Quantifying The Effect of Wakefield et al. (1998) on Skepticism about MMR Vaccine Safety in the U.S.

PLOS ONE

Dear Dr. Motta,

Thank you for submitting your manuscript to PLOS ONE. After careful consideration, we feel that it has merit but does not fully meet PLOS ONE’s publication criteria as it currently stands. Therefore, we invite you to submit a revised version of the manuscript that addresses the points raised during the review process.

First of all, apologies for the excessive length of this round of reviews; it has been especially difficult to locate available reviewers to asses such a specific paper. Once it has been reviewer by three experts, I would like to ask you for a set of revisions (most of them relatively minor), even though major issues such as the potential role of confounding variables and a complementary analysis suggested by Reviewer # 2 should be also responded by you. Once received your resubmission, and as the reviewers are already available and willing to view your revised paper, I will try to expedite the processing times as much as possible.

We look forward to receiving your revised manuscript.

Kind regards,

Sergio A. Useche, Ph.D.

Academic Editor

PLOS ONE

Journal Requirements:

Reviewers' comments:

Reviewer's Responses to Questions

**Comments to the Author**

1. Is the manuscript technically sound, and do the data support the conclusions?

Reviewer #1: Yes

Reviewer #2: Yes

Reviewer #3: Partly

2. Has the statistical analysis been performed appropriately and rigorously? 

Reviewer #1: Yes

Reviewer #2: Yes

Reviewer #3: Yes

3. Have the authors made all data underlying the findings in their manuscript fully available?

Reviewer #1: Yes

Reviewer #2: Yes

Reviewer #3: Yes

4. Is the manuscript presented in an intelligible fashion and written in standard English?

Reviewer #1: Yes

Reviewer #2: Yes

Reviewer #3: Yes

5. Review Comments to the Author

Reviewer #1: This is a manuscript examining media coverage of MMR vaccine in the time frame around the 1998 publication of the infamous Wakefield Lancet paper and examining MMR-related VAERS claims before and after the publication of that paper. Overall this is interesting and contributes to our understanding of vaccine hesitancy, and is particularly timely given the situation with COVID vaccination in 2021.

There is one thing in the manuscript that needs to be clarified. The authors use PPV (pneumococcal polysaccharide vaccine) as a control for time as a variable, showing the data in Figure S1. This figure and the underlying analysis is problematic. PPV is not routinely administered to children, but pneumococcal conjugate vaccine (PCV) is. PPV is typically given to older adults at risk of pneumococcal pneumonia, not children at risk for pneumococcal otitis and sinusitis. Prior to the availability of PCV, PPV was given to children at high risk of pneumococcal disease (eg, children with sickle cell disease, children who lacked splenic function), but the number of immunized children was small. PCV was licensed in the US in February 2000 and was not widely available initially. In addition, there was a slight delay in uptake of the vaccine as insurers waited for efficacy data before agreeing to pay for a new, somewhat expensive vaccine. The rise in claims starting with the increase in 2000 and the later sharper rise around 2004 (as shown in Figure S1) is consistent with the uptake of PCV. Since these data are almost certainly a concatenation of PPV and PCV claims, this needs to be clarified in the manuscript and figure legend. That being said, analysis of Haemophilus influenzae type b conjugate vaccine data would not have this temporal confounding since that vaccine was introduced in the early 1990s. I would not require that this additional analysis be done, but do note that the data would be "cleaner".

Reviewer #2: Considering the slow vaccination rate, in the Discussion the authors should extend their comments regarding the implications of their work on the Corona vaccine skepticism. They should point out to Corona related misinformation similar to the MMR misinformation

Reviewer #3: The authors underscored an actual subject, unfound concerns on vaccine safety, by recalling a landmark event like the spurious publication by Wakefield et. al. On this task, the authors traced back claims on VAERS along three decades (1990-2019). The use of another vaccine as a control provided an interesting approach to assess the data. Nonetheless, three decades is a significant amount of time and during the period, several changes have occurred including changes in reporting systems and awareness of autism diagnosis. Regarding the first aspect, the use of the "control" vaccine partly answered this issue. But, changes in the way to access the reporting system should be considered also because they can affect the overall reporting. It is also advisable to  put the number of claims in the perspective of the number of vaccinees per year as denominators to understand the data. On the autism diagnosis aspect, the changes along the study period are outstanding (see https://www.ncbi.nlm.nih.gov/books/NBK332896/).  Autism was largely unknown in 1990's and awareness of healthcare professionals and the public might play a role in the apparent increase in the diagnosis (see BMJ, 312 (1996), pp. 327-328). The media coverage on the Wakefield's false claim might also raise awareness of autism diagnosis during the period (see Ment Retard Dev Disabil Res Rev 2002;8(3):151-61). In summary, the idea proposed by the authors has relevance and an interesting approach, but confounding factors should be better addressed to build stronger analyses of the data.

6. PLOS authors have the option to publish the peer review history of their article (what does this mean?). If published, this will include your full peer review and any attached files.

Reviewer #1: No

Reviewer #2: **Yes: **Michael Davidson MD

Reviewer #3: No

---

## [Author Response · Author response to Decision Letter 0]

22 Jul 2021

** Please see attached Word Document reply. (Comments re-printed below):

Dear Dr. Useche & Reviewers at PLOS ONE,

First, I’d like to sincerely thank you for taking the time to review our manuscript entitled Quantifying The Effect of Wakefield et al. (1998) on Skepticism about MMR Vaccine Safety in the U.S.. Dom and I truly appreciate your willingness to provide feedback on our work, especially during these challenging and unprecedented times.

We were delighted to hear that the piece received a revision decision, and found the reviewers’ comments to be very helpful in revising our manuscript. Although Dr. Useche noted that many of the revisions would require only minor changes to the manuscript, we have nevertheless made several important changes to the updated piece.

Before offering a point-by-point reply, we wanted to briefly summarize some of the more-major changes we made to the updated manuscript. Chiefly, and pursuant with the Reviewers’ recommendations, we have:

1. Provided updated placebo analyses that swap reports to VAERS regarding PPV for a “cleaner” test (HIBV). [As suggested by Reviewer 1]

2. Added a discussion of how our work relates to the ongoing COVID-19 vaccination effort. [As suggested by Reviewers 1 and 2] 

3. Described how it is that our placebo tests can guard against confounds not originally referenced in the main text. [As suggested by Reviewer 3]

We have also made a series of more minor edits throughout the manuscript, pursuant with both Dr. Useche’s comments and those of Reviewers 1-3.

Below, we reprint each comment raised by Reviewers 1-3 (in bold) and list our replies below each one. Where relevant, we also include block text quotes from the manuscript, to more-easily document the changes we made.

We thank you all again for your time and effort, and look forward to continuing to work with you throughout the review process.

Sincerely,

Matt Motta & Dom Stecula

REVIEWER 1 (R1)

This is a manuscript examining media coverage of MMR vaccine in the time frame around the 1998 publication of the infamous Wakefield Lancet paper and examining MMR-related VAERS claims before and after the publication of that paper. Overall this is interesting and contributes to our understanding of vaccine hesitancy, and is particularly timely given the situation with COVID vaccination in 2021.

We thank R1 for their kind comments about our manuscript, and noting its application to COVID-19 vaccine uptake in 2021. We also wanted to briefly note here that we now expand on relevance of our work to COVID-19 vaccination in response to comments offered by R2. That discussion can be found in the Discussion section of the main text, and is printed in full in our reply to R2.

There is one thing in the manuscript that needs to be clarified. The authors use PPV (pneumococcal polysaccharide vaccine) as a control for time as a variable, showing the data in Figure S1. This figure and the underlying analysis is problematic. PPV is not routinely administered to children, but pneumococcal conjugate vaccine (PCV) is. PPV is typically given to older adults at risk of pneumococcal pneumonia, not children at risk for pneumococcal otitis and sinusitis. Prior to the availability of PCV, PPV was given to children at high risk of pneumococcal disease (eg, children with sickle cell disease, children who lacked splenic function), but the number of immunized children was small. PCV was licensed in the US in February 2000 and was not widely available initially. In addition, there was a slight delay in uptake of the vaccine as insurers waited for efficacy data before agreeing to pay for a new, somewhat expensive vaccine. The rise in claims starting with the increase in 2000 and the later sharper rise around 2004 (as shown in Figure S1) is consistent with the uptake of PCV. Since these data are almost certainly a concatenation of PPV and PCV claims, this needs to be clarified in the manuscript and figure legend. That being said, analysis of Haemophilus influenzae type b conjugate vaccine data would not have this temporal confounding since that vaccine was introduced in the early 1990s. I would not require that this additional analysis be done, but do note that the data would be "cleaner".

R1 goes on to note that the use of the PPV vaccine as a placebo in our analyses, while not necessarily inappropriate, is nevertheless complicated by (1) its selective administration throughout the 1990s, and (2) the introduction of PCV in 2000. We think that this is a very important point, and thank R1 for bringing their subject matter expertise to bear on this question. 

Pursuant with R1’s recommendations, we have swapped the PPV placebo test for the reviewer’s recommended alternative: HIBV. We completely agree with R1 that this offers a “cleaner” test of our expectations. Indeed, as we now report in Figure S1, our placebo test results again document no clear statistical or substantive effect of AW98 on reports to VAERS for the HIB vaccine. 

We also wanted to provide a brief and somewhat technical note about these updated placebo analyses. Due to relatively high levels of VAERS reports logged at the start of the time series (i.e., when HIBV was first introduced), we observe an inverted parabolic trend in VAERS reports prior to 1998 (high at the start of the series, falling, and then increasing years prior to AW98). To pose the fairest possible test for spuriousness, we therefore account for the nonlinear nature of the pre-AW98 data by using quadratic (as opposed to linear) regression when estimating our bias correction RD models. These analyses again document no increase in VAERS reports following AW98 (in fact, once we account for the quadratic nature of the data, we observe a very slight decrease in reports).

- 

REVIEWER #2 (R2)

Reviewer #2: Considering the slow vaccination rate, in the Discussion the authors should extend their comments regarding the implications of their work on the Corona vaccine skepticism. They should point out to Corona related misinformation similar to the MMR misinformation

R2’s primary concern with the manuscript is that we did not make a sufficient effort to engage with COVID-19 vaccine uptake in the piece’s discussion section. We completely agree with R2’s comments, on this score, and note that R1 mentioned this point in passing as well.

Correspondingly, we have added two paragraphs of text to the piece’s discussion section wherein we discuss the potential applications of our research for understanding vaccine uptake throughout the global pandemic. We also point to applications for future research, and (briefly) sketch out VAERS-related studies that researchers might consider taking up. 

Specifically, we write: 

“Additionally, and perhaps most importantly, our research offers an opportunity to learn more about the effects of COVID-19 vaccine media coverage on attitudes toward vaccination. As more Americans became eligible to vaccinate against the virus, vaccine skeptics and prominent media figures who held skeptical views toward vaccination sometimes made use of VAERS reports to justify their (often, misinterpreted) concerns about COVID-19 vaccine safety [20-22]. Consequently, future work should make an effort to determine whether or not media coverage of these issues in turn influenced reports to VAERS. This information can help scholars to not only better understand the effects of media coverage on vaccine attitudes, in application to diseases not studied in our work, but how the quality of adverse effect data reported to VAERS might be influenced by exogenous media, political, and other social events.

Moreover, although public opinion data about COVID-19 vaccine attitudes and behaviors is comparatively more plentiful than MMR opinion data, few surveys ask uptake questions at the daily level. As we demonstrate in this research, VAERS data – which can be readily aggregated at the daily level – could function as a useful supplement for public opinion data to detect short-term effects of changes in the vaccine communication environment. For example, researchers could use VAERS data to document the effects of media attention the federal government’s decision to temporarily pause administration of Johnson & Johnson’s COVID-19 vaccine. Because this pause lasted just ten days [23], VAERS data may offer an opportunity to assess these effects with additional granularity.”

We hope that these remarks help broaden the scope of the relevance of our work to current events, and facilitate additional research using these under-studied data.

REVIEWER #3 (R3)

The authors underscored an actual subject, unfound concerns on vaccine safety, by recalling a landmark event like the spurious publication by Wakefield et. al. On this task, the authors traced back claims on VAERS along three decades (1990-2019). The use of another vaccine as a control provided an interesting approach to assess the data. Nonetheless, three decades is a significant amount of time and during the period, several changes have occurred including changes in reporting systems and awareness of autism diagnosis. Regarding the first aspect, the use of the "control" vaccine partly answered this issue. But, changes in the way to access the reporting system should be considered also because they can affect the overall reporting. 

R3 begins by noting that, while our study’s placebo vaccine analyses can help buttress concerns that differences in how health care providers diagnose autism might influence the effects of AW98 observed in the main text, it is also important that we consider the possibility that changes in the mechanics of submitting reports to VAERS might have a potentially confounding effect.

We completely agree with R3 on this score, and have made several changes to the main text in response. As we now note directly in the main text (amended language re-printed below), we believe that the placebo tests can also guard against this concern. If it was the case that changes in the reporting system were to (say) co-occur with AW98 and therefore be responsible for increased reporting, we would also expect to see similar increases in reports on other vaccines (as reporting changes, like the opportunity to report claims online, are VAERS-level phenomena, and not necessarily specific to one vaccine). 

However, we recognize that we did not make this point clearly in the initial manuscript. Consequently, we now list several potential ways in which our results could be confounded by events aside from AW98, and include language suggesting that the placebo tests are well suited to address these concerns. Specifically, we write (updated text in bold): 

“Additionally, to ensure that growth MMR concern is not confounded by (1) changes in the procedures by which parents report adverse effects to VAERS over time (e.g., the ability to submit reports online as personal home computing and internet access grew over time), (2) changes in how health care providers diagnose autism, and/or (3) time itself – i.e., the possibility that attitudes toward all vaccines grew more negative over time for reasons unrelated to AW98 – we offer a placebo test using reports of adverse reactions to HIBV (Haemophilus Influenzae Type B Vaccine) – which has been routinely administered to children since the early 1990s, but was not identified as a health risk in AW98 – in the online materials.

If the effects of AW98 were to coincide with unobserved changes in reporting mechanisms and/or more general changes in public vaccine sentiment, we should observe comparatively more VAERS reports in the post (vs. pre) AW98 period for the HIBV placebo. This would be indicative of a spurious effect of AW98. However, the analyses presented in the online materials suggest that AW98 had no statistically or substantively discernable effect on HIBV adverse event reports (see: S1 Figure).”

It is also advisable to put the number of claims in the perspective of the number of vaccinees per year as denominators to understand the data. 

R3 goes on to note that population-adjusted vaccine estimates (i.e., the percentage VAERS reports relative to all MMR doses administered) might help some readers to better engage with the analyses presented in the main text. We think that this is a thoughtful point, and one that we took seriously when revising this piece. We have two brief replies.

First, we wanted to underscore that – although we present the raw number of reports to VAERS in Figure 1 – we do display a model in the figure that adjusts for changes in US population growth over time. Our results are robust to this alternative specification.

Second, we nevertheless recognize that the quantities R3 suggests may be more tractable for some readers. Unfortunately, available time serial MMR surveillance data from the federal government are typically expressed in percentages, and not raw counts, throughout the period we study. While we could theoretically multiply the size of the vaccine-eligible population by these percentages, doing so would introduce additional error into our analyses – as percentages are necessarily rounded (so multiplying one by the other would imply a loss of precision). 

(Note: we are of course open to the possibility that these data are readily available, and would be happy to consider re-estimating the models if we are able to access the data).

Still, we note that MMR vaccine uptake rates have held relatively constant throughout the series that we study in this piece. This likely means that the results we observe when we correct for population change in our RD models would produce a similar result to scaling VAERS reports by the total number of vaccines administered (as the population totals would be multiplied by a more-or-less constant factor). 

Consequently, though we have not amended our approach in the main text to reflect this concern, we nevertheless think that it is a thoughtful point, and wanted to ensure that we gave it significant consideration in our reply memo.

On the autism diagnosis aspect, the changes along the study period are outstanding (see https://www.ncbi.nlm.nih.gov/books/NBK332896/). Autism was largely unknown in 1990's and awareness of healthcare professionals and the public might play a role in the apparent increase in the diagnosis (see BMJ, 312 (1996), pp. 327-328). The media coverage on the Wakefield's false claim might also raise awareness of autism diagnosis during the period (see Ment Retard Dev Disabil Res Rev 2002;8(3):151-61). 

Finally, R3 suggests that changes in the diagnosis of autism throughout the study period might complicate our account of the potential mechanism for the effects we observe in this piece. Perhaps media attention to AW98 raised awareness about autism, which in turn altered the way in which health care providers make diagnoses? 

We think that this is an important point, and one we now discuss directly in the manuscript’s main text. This alternative account, in our view, adds a level of conceptual nuance to our proposed mechanism (i.e., negative media coverage of the MMR vaccine). In other words, it suggests that media coverage has downstream effects on how doctors and patients interact, which in turn influence VAERS reports. Media coverage remains the “starting point” of the causal chain, under this view, but takes more-nuanced “turns” before ultimately influencing reports.

As we now discuss in the main text (remarks printed below; note also the inclusion of the helpful Wing & Potter piece that R3 recommended we consult), we think that this is a fascinating direction for future research. Although we characterized our discussion of the hypothesized mechanism for the effects observed in this piece as preliminary in our initial manuscript, we encourage scholars in the future to explore these more-nuanced causal accounts. In fact, we are actively trying to do this ourselves, in working research. 

Specifically, we write:

“Finally, our work suggests several opportunities for future research. For example, our study cannot disentangle the precise mechanism by which AW98 media coverage might influence public vaccine attitudes. Media attention could, for example, directly influence how Americans feel about vaccination by drawing parents’ attentions to the possibility that their children experience adverse side effects from vaccination. Alternatively, the vaccine media environment could indirectly influence vaccine attitudes indirectly by changing health care professionals attentiveness to potential side effects (e.g., increased monitoring for autism symptoms), and thereby engender concern among parents resulting in increased VAERS reports. Efforts to disentangle direct from more-complex causal accounts are a worthwhile endeavor in previous research.”

In summary, the idea proposed by the authors has relevance and an interesting approach, but confounding factors should be better addressed to build stronger analyses of the data.

We thank R3 for their kind remarks. We hope that these changes show that we have taken seriously the ways in which this manuscript accounts for potential confounding effects, and made an effort to clarify our efforts to address this issue.

---

## [Decision Letter · Decision Letter 1]

6 Aug 2021

Quantifying The Effect of Wakefield et al. (1998) on Skepticism about MMR Vaccine Safety in the U.S.

PONE-D-21-02863R1

Dear Dr. Motta,

We’re pleased to inform you that your manuscript has been judged scientifically suitable for publication and will be formally accepted for publication once it meets all outstanding technical requirements.

Kind regards,

Sergio A. Useche, Ph.D.

Academic Editor

PLOS ONE

Additional Editor Comments (optional): Thanks for your revisions and clarifications (and apologies for the delay).

Reviewers' comments:

Reviewer's Responses to Questions

**Comments to the Author**

1. If the authors have adequately addressed your comments raised in a previous round of review and you feel that this manuscript is now acceptable for publication, you may indicate that here to bypass the “Comments to the Author” section, enter your conflict of interest statement in the “Confidential to Editor” section, and submit your "Accept" recommendation.

Reviewer #1: All comments have been addressed

Reviewer #2: All comments have been addressed

Reviewer #3: All comments have been addressed

2. Is the manuscript technically sound, and do the data support the conclusions?

Reviewer #1: Yes

Reviewer #2: Yes

Reviewer #3: Yes

3. Has the statistical analysis been performed appropriately and rigorously? 

Reviewer #1: Yes

Reviewer #2: Yes

Reviewer #3: Yes

4. Have the authors made all data underlying the findings in their manuscript fully available?

Reviewer #1: Yes

Reviewer #2: Yes

Reviewer #3: Yes

5. Is the manuscript presented in an intelligible fashion and written in standard English?

Reviewer #1: Yes

Reviewer #2: Yes

Reviewer #3: Yes

6. Review Comments to the Author

Reviewer #1: The authors have addressed my concerns, and I am particularly pleased that they took the suggestion to look at an alternative vaccine comparator.

I have a few comments for copyediting, but none are problematic.

Line 97: The work “taking” seems out of place.

Line 99: I believe the sentence starting, “In this study,” is missing a subject. It seems like it should read, “In this study, we make use…”.

Line 218: I believe the sentence is missing a preposition in, “to ensure that growth MMR concern”.

Reviewer #2: As requested the authors commented in the discussion section on the relevance of their work to Corona hesitancy All remarks have been addressed. No additional revision are necessary.

Reviewer #3: After the review, the manuscript is more clear and properly addressed potential cofounders in the text. Switching the "control" vaccine to HiBV helped to validate results in better than in the previous version. Besides the discussion on the cofounders cannot be resolved in retrospective way, the authors acknowledged this limitation.

One last comment is to suggest using "control test" instead of "placebo test" referring to the HiBV data. The purpose of this test is to offer a comparison and not to pretend to be the evaluated vaccine.

Congratulations to the authors for the interesting manuscript.

7. PLOS authors have the option to publish the peer review history of their article (what does this mean?). If published, this will include your full peer review and any attached files.

Reviewer #1: **Yes: **Michael Anthony Moody

Reviewer #2: **Yes: **Michael Davidson

Reviewer #3: No

---

## [Editor Report · Acceptance letter]

10 Aug 2021

PONE-D-21-02863R1 

Quantifying The Effect of Wakefield et al. (1998) on Skepticism about MMR Vaccine Safety in the U.S. 

Dear Dr. Motta:

I'm pleased to inform you that your manuscript has been deemed suitable for publication in PLOS ONE. Congratulations! Your manuscript is now with our production department. 

Kind regards, 

on behalf of

Dr. Sergio A. Useche 

Academic Editor

PLOS ONE